# ScanBious: Survey for Obesity Genes Using PubMed Abstracts and DisGeNET

**DOI:** 10.3390/jpm11040246

**Published:** 2021-03-29

**Authors:** Svetlana Tarbeeva, Ekaterina Lyamtseva, Andrey Lisitsa, Anna Kozlova, Elena Ponomarenko, Ekaterina Ilgisonis

**Affiliations:** 1International School “Medicine of the Future”, Sechenov University, 119991 Moscow, Russia; 2Institute of Biomedical Chemistry, 119121 Moscow, Russia; lisitsa060@gmail.com (A.L.); ministreliya13113@gmail.com (A.K.); 2463731@gmail.com (E.P.); ilgisonis.ev@gmail.com (E.I.); 3Institute of Biology, Tyumen State University, 625003 Tyumen, Russia; 24ket2001@mail.ru; 4Laboratory of Molecular Design and Synthesis, Shemyakin-Ovchinnikov Institute of Bioorganic Chemistry of the Russian Academy of Sciences, 117997 Moscow, Russia

**Keywords:** gene network, obesity, text-mining, data-mining, MeSH

## Abstract

We used automatic text-mining of PubMed abstracts of papers related to obesity, with the aim of revealing that the information used in abstracts reflects the current understanding and key concepts of this widely explored problem. We compared expert data from DisGeNET to the results of an automated MeSH (Medical Subject Heading) search, which was performed by the ScanBious web tool. The analysis provided an overview of the obesity field, highlighting major trends such as physiological conditions, age, and diet, as well as key well-studied genes, such as adiponectin and its receptor. By intersecting the DisGeNET knowledge with the ScanBious results, we deciphered four clusters of obesity-related genes. An initial set of 100+ thousand abstracts and 622 genes was reduced to 19 genes, distributed among just a few groups: heredity, inflammation, intercellular signaling, and cancer. Rapid profiling of articles could drive personalized medicine: if the disease signs of a particular person were superimposed on a general network, then it would be possible to understand which are non-specific (observed in cohorts and, therefore, most likely have known treatment solutions) and which are less investigated, and probably represent a personalized case.

## 1. Introduction

It is not difficult to diagnose obesity using the body mass index. Unlike other systemic diseases, obesity is reversible. Therefore, it is an interesting task to predict, based on molecular analysis, the personal risk of obesity in the future. Additionally, prediction is an opportunity to reverse the process with minor lifestyle changes. An analysis of the current level of knowledge accumulated in the form of scientific articles is a starting point for defining the molecular mechanisms of the development of obesity. In the future, semantic maps of the subject area will be able to be compared with a person’s molecular profile, for the development of molecular patterns and personalized patterns, affecting lifestyle. Comparison of a molecular map using a domain concept is required to find trends and to plan further research.

A superficial analysis of publications on the topic of obesity reveals a “hit parade” of genes, hormones, proteins and metabolites that are frequently mentioned in articles. These include insulin (as a glucose regulator), adiponectin (as a protein synthesized by adipose tissue), the inflammatory cascade, and obesity as a typical homeostasis disorder.

In 2012, a review was published [1] summarizing 15 years of research in the genetics of obesity. It examined the existence of a genetic relationship between monogenic and polygenic obesity. The roles of genes involved in the regulation of food intake in terms of the characteristics of the nervous system and genetic predisposition to obesity are discussed.

Obesity is a disease which includes two major ways: The first is metabolic dysregulation and the second is dysregulation of the lipid metabolism. In this paper we were interested in whether these interconnected mechanisms can be derived by the automated processing of the scientific texts.

The semantic presentation of molecular diseases was proposed by Professor Barabasi in one of his earliest works, which dates back to 2007 [2,3]. It showed that the analysis of a sample of articles can be represented as a network, where the nodes are the concepts—for example, genes and diseases. This approach has not become obsolete: for example, automatic analysis of the texts of scientific publications has been applied to investigate the problem of Coronavirus Disease 2019 (COVID-19) [4]. Earlier, bioinformatic scientists from China showed the possibility of using GoPubMed analysis to handle the boom in publications related to CRISPR/Cas9 [5]. In this work and subsequent studies [6,7], the information contained in the abstract or in the full text of the article was shown to be sufficient to establish relationships between molecules that are important for the functioning of living systems. Professor Barabasi’s approaches have been developed into a number of information and analytical systems, including GoPubMed [8] and the BiblioEngine package [9].

The effectiveness of text mining by using the PubMed as a base engine was illustrated in iTextMine [10]. This system allows you to display the relationships between genes, proteins, including kinase enzymes, miRNAs, diseases, medications and responses to pharmacotherapy. 

A comparison of the results of automatic extraction of gene names from abstracts of publications with the results of expert analysis of publications showed that there is a loss of information [11]. For the example of obesity, this was done using the obesity and co-morbid diseases database (OCDD) [12]. However, the junction between expert accumulated knowledge and automated information processing could be focused on diseases of social importance, obesity among them.

We used DisGeNET [13] as a platform to integrate information of human gene-disease associations from various repositories, including Mendelian, complex, and environmental diseases. The DisGeNET knowledgebase allows user retrieval of gene-disease associations, referencing the PubMed identifiers of those articles from which the fact was derived by an expert. It integrates studies of dietary, genetic, physiological, and psychological/behavioral factors [13]. It is important to emphasize that experts enter data into DisGeNET based on full-text articles, not just abstracts.

The purpose of our work was to reveal key concepts and trends in the field of human obesity studies, based on compared data from DisGeNET and the results of automatic processing of PubMed/MEDLINE data. The information in this article may help in the study of poorly understood factors of obesity, by focusing on neglected genes.

During the annotation process, PubMed curators assign abstracts with keywords; the so-called Medical Subject Headings (MeSH). MeSHs are used to structure knowledge; see the work of Gan et al. on epilepsy for an example [14].

## 2. Materials and Methods

### 2.1. Jaccard Index for PubMed Abstracts

For the given MeSH terms or gene names, a sample of publications was retrieved from either DisGeNET or PubMed. Automatic loading of publications from PubMed, their abstracts, MeSH terms, and analysis of the frequency of occurrence of MeSH terms, as well as visualization of connections between them, were performed using the ScanBious web tool (https://scanbious.ru/, accessed on 26 March 2021).

In ScanBious, the construction of relationships between objects relies on a transparent notation of the Jaccard index, similarity measure applied to the binary intersections (also known as Tanimoto measure) [9]. The Jaccard index describes the degree of similarity between the two sets; for example, the sets of Pubmed IDs (PMIDs) of papers in which an MeSH term or gene name occurs, by the formula:K = C/(A + B − C),(1)
where K is the Jaccard index (values from 0 to 1; the closer to 1, the more similar the sets are), A and B are the numbers of PMIDs in the two sets, and C is the number of common PMIDs in the two sets.

### 2.2. ScanBious Interface

ScanBious was developed as a free-ware Web-system for highlighting key concepts revealed from PubMed abstracts and related MeSH terms. In contrast to the deep-learning instruments for scientific text mining, ScanBious provides an interface to the transparent algorithm, which relies on the co-occurrence of two terms in one abstract.

ScanBious is a system, the functionality of which includes sorting by frequency of occurrence of key concepts and research objects in a given subject area.

ScanBious provides the user an opportunity to concentrate on events that are frequently researched and therefore represented in many publications. Vice versa—to select from the huge array of information the specific, little-studied events, or new hypotheses or unobvious interplay for further research. Since the distribution of keywords such as MeSH terms or gene names, follows Zipf’s law, the principle of the ScanBious search facts can be called “information depletion,” when the semantic map of the explored area is iteratively cleaned up from well-known nodes or superficial relationships.

The general scenario of working with the ScanBious system is shown in Figure 1. At the request of the user (I), the system loads abstracts and MeSH terms corresponding to the request found in the PubMed/MEDLINE system. The user selects objects (II), for which the measure of interrelation (Jaccard index) is calculated and the semantic network (IV) is visualized, the nodes of which are the objects selected by the user, and the links of which are a list of publications in which both objects are described. The functionality of the system provides for the ability of the user to refer to the texts of abstracts of relevant publications for the object (issued by clicking on the semantic map node, V) and the texts of abstracts of publications substantiating the existence of a relationship between objects. When working with a semantic map, the functionality of the ScanBious system provides for the ability to save fragments of abstracts in the user’s personal account for use in the user’s further work; for example, when writing literary reviews (V).

### 2.3. Resources and Workflow

To implement this work, we used the current (September 2020) versions of three resources: UniProt as a source of gene names, DisGeNET as a source of gene names associated with obesity, and PubMed as a source of abstracts.

The workflow was based on the intersection of the sets, uploaded from the aforementioned resources. We denote PP as the set of articles from the Publish or Perish resource [15], UP as the set of gene names retrieved from the human UniProt entries, DGN as the set of gene names from the DisGeNET portal, and PM as the set of identifiers of the PubMed abstracts. Taking these notations, we can formally express our workflow as a pseudocode of consecutive intersections (symbol ‘$\cap^{}$’) of the sets:PP $\cap^{}$ “obesity” = PPOb → MeSH NetworkPM $\cap^{}$ “obesity” = PMObPMOb $\cap^{}$ UP = UPObDGN $\cap^{}$ “obesity” = DGNOb → Gene NetworkUPOb $\cap^{}$ DSNOb → Gene Clusters

Firstly, an overview of the problem is made by referring to the most cited authors, for step 1. At step 2 the obesity relevant PubMed abstracts, which are matched to the UniProt gene identifiers, are selected for step 3. The search for obesity-related genes is also undertaken using the DisGeNET interface at step 4. At step 5, two sets of genes are intersected with each other to dissect the scope of the obesity problem in the clusters.

## 3. Results

### 3.1. Obesity Overview

A ScanBious-produced basic representation of papers related to the obesity problem is shown in Figure 2a as a network of MeSH terms, while Figure 2b shows the network relationships between genes. Figure 2a shows a network of MeSH terms built on the basis of automatic analysis of 100+ thousand articles on the molecular mechanisms of obesity, published in the last ten years. The node size corresponds to the number of publications. Risk factors, body mass index, childhood, preschool and adolescence, and the epidemiology of obesity are the largest nodes. Large nodes determine, in particular, the developmental characteristics (adolescent, child), habits, gender, and corresponding constitutional differences. Other factors are distributed across smaller nodes. Obesity is associated with a specific phenotype (risk factors, diet) and with specific physiological conditions, such as pregnancy, growing up and sex.

### 3.2. Network of Obesity Genes from DisGeNET

Figure 2b shows a gene network obtained using a co-occurrence analysis of gene names in articles retrieved from DisGeNET for obesity. From DisGeNET, we selected 13,768 gene-obesity associations for 2710 human genes, according to the data from 9169 papers. In the figure, large nodes are observed corresponding to adiponectin, leptin and its receptor, tumor necrosis factor, and C-reactive protein, as well as peroxisome proliferator-activated receptor gamma (PPRAG) with its co-activator 1-alpha (PPRAGC1A). The size of the nodes reflects the number of publications in which the gene name was found. For example, adiponectin has been studied for over 40 years and was referred to in 418 publications according to Figure 2b. It is a hormone encoded by the ADIPOQ gene and synthesized and secreted by white adipose tissue, predominantly by adipocytes of the visceral region [16]. Adiponectin is involved in the regulation of glucose levels and the breakdown of fatty acids [17].

The two dominant nodes in Figure 2b correspond to leptin (LEP) and leptin receptor (LEPR, mentioned in 244 papers in the set). In the hypothalamus, leptin acts as an appetite-regulating factor by reducing food intake and increasing energy intake, by inducing anorexigenic factors and suppressing orexigenic neuropeptides.

The next most commonly mentioned gene is tumor necrosis factor, (TNF). This factor is responsible for insulin resistance in adipocytes, in conjunction with blood insulin levels. TNF-related publications have supported that obesity is an inflammatory process. In addition to TNF-α, it is characterized by increased production of pro-inflammatory cytokines such as interleukins (IL6 and IL18) and C-reactive protein (Figure 2b). Thus, by examining the publications associated with the largest nodes using the ScanBious functionality, we get a general idea of the problem in semantic coordinates: (1) white fat as a source of secreted bioactive substances, (2) importance of the neurogenic factors in obesity [18], and finally (3) TNF-associated inflammation processes.

Large nodes of the semantic gene network in Figure 2b generally reflect well known facts, unlike the smallest nodes; for example, CPE (Carboxypeptidase E), NISCH (Nischarin) and ZBTB7C (Zinc finger and BTB domain containing 7C). For the first two genes listed, information on function is contained in the UniProt database: CPE, or carboxypeptidase E, directs prohormones to the regulated secretory pathway, while NISCH is a multifunctional protein, responsible for initiation of a wide range of cellular signaling cascades. For the third gene, zinc finger and BTB domain-containing protein 7C (ZBTB7C), UniProt contains the entry “May be a tumor suppressor gene.” Therefore, both the large node for tumor necrosis factor and one of the small nodes for ZBTB7C indicate the global semantic axis in obesity research: cancer, which is a frequent companion of inflammation [19].

### 3.3. Genetic Determinants of Obesity

When considering Figure 2b, we can see the absence of genes in which mutations are associated with obesity. Information about these associations was obtained from GWAS research. Choquet, 2012 [1] presented information on eight monogenic genes and four polygenic genes, but none of these polygenic genes are shown in Figure 2b. Bauer et al. [20] reported evidence for an association of additional obesity genes identified by GWAS (SH2B1, KCTD15, MTCH2, NEGR1, BDNF) with dietary intake and nutrient-specific food preferences. None of these genes are observed in Figure 2b, nor are other genes mentioned in the seminal update of the obesity genetics basics [21].

Most of the Mendelian-inherited genes were presented in the results of the DisGeNET and PubMed search. In Table 1, we see the most abundant examples of obesity relevant genes. Most of them (four out of five) failed to pass the Jaccard filter during the construction of the network in Figure 2b. This suggests that non-evident but statistically proven (e.g., as the odds ratio in GWAS) facts are missing from the semantic scheme. This could be explained by the poor links between obesity-involved gene mutations and other genes, which may participate in the multifactorial mechanisms of obesity. For instance, for FTO alpha-ketoglutarate dependent dioxygenase (FTO) we observed 426 PubMed entries in DisGeNET; half as many PubMed identifiers (PMIDs) were retrieved for the leptine receptor (LEPR). Due to the higher degree of interconnections with other genes, LEPR was presented in the network (Figure 2b), while the more highly cited gene FTO was missing.

### 3.4. Combining the PubMed Survey with DisGeNET Data on Obesity Genes

The data from DisGeNET was combined with the ability of ScanBious to process and represent the information from PubMed to reduce the complexity of the published facts about obesity. Dissecting the results at a Jaccard index value above 0.2, we obtained just four clusters, shown in Figure 3. The number of genes in a cluster ranged from 3 to 8, and there were no complex hairballs of links between the nodes; these are a curse of the network approach in biology.

Cluster A formed a network of eight genes: INPP5E, VPS13B, FBN3, HECTD4, BBS12, BBS10, BBS7, and WDR11. The largest number of publications, 87, was observed for INPP5E (phosphatidylinositol polyphosphate 5-phosphatase type IV). This phosphatase plays a role in the primary cilium by controlling ciliary growth and phosphoinositide 3-kinase signaling and stability.

The next participant of Cluster A, the VPS13B gene, encodes the vacuolar protein sorting-associated protein 13B. With an autosomal-recessive mutation on chromosome 8q22.2, this gene can cause Cohen’s syndrome, which is characterized by obesity, hypotension, intellectual disabilities, characteristic craniofacial dysmorphism, and abnormalities in the development of the hands and feet [22].

FBN3 gene polymorphism (Fibrillin-3) regulates the activity of transforming growth factor-b (TGF-b) and regulatory levels of T-cells [23]. This implies a link between this gene and inflammatory processes. For the next gene in this cluster, encoding probable E3 ubiquitin-protein ligase (HECTD4), no information about its function has been found, making it a promising target for obesity research.

The set of genes associated with Bardet-Biedl syndrome is represented in cluster A by the nodes BBS12, BBS10, and BBS7. These are components of the chaperonin-containing T-complex (TRiC), a molecular chaperone complex that assists with the folding of proteins. This part of the TRiC complex plays a role in the assembly of BBSome, a complex involved in ciliogenesis regulating transport vesicles to the cilia [24]. It is also involved in adipogenic differentiation [25]. Another member of Cluster A is also essential to ciliogenesis. WDR11, the WD repeat-containing protein 11, is involved in the Hedgehog (Hh) signaling pathway [26]. It regulates the proteolytic processing of zinc finger protein GLI3, and cooperates with the transcription factor EMX1 in the induction of downstream Hh pathway gene expression and gonadotropin-releasing hormone production [26]. WDR11 complex facilitates the tethering of vesicles produced with the adaptor protein-1 (AP-1). WDR11-regulated assembly acts in consortia with TBC1D23, and invokes the capture of vesicles generated by AP-1 [27].

In cluster B, we observe that CHRNA2 (cholinergic receptor nicotinic alpha 2) is expressed at significant levels in subcutaneous adipocytes [28], and one of its forms might be a risk factor for obesity in Koreans [29]. GRID2 (glutamate receptor, ionotropic, delta 2) is selectively expressed in Purkinje cells in the cerebellum, and at first glance is weakly associated with obesity. However, GRID2 was identified as the most likely candidate gene within the body weight locus of the human genome. It is of interest that several of the candidate genes of that locus play a role in neural regulation of energy metabolism and feeding behavior [30].

Two other inhabitants of cluster B, FAIM2 and KCTD15, appeared to be guilty-by-association with obesity, while SMG6 (telomerase-binding protein EST1A) is a ubiquitously expressed enzyme with no significant evidence for having an effect on the development of obesity. Therefore, FAIM2 (Fas apoptotic inhibitory molecule 2) is an anti-apoptotic protein which protects cells uniquely from Fas-induced apoptosis. It regulates Fas-mediated apoptosis in neurons by interfering with caspase-8 activation. It may play a role in cerebellar development by affecting cerebellar size, internal granular layer thickness, and Purkinje cell development [31,32]. There is a correlation between the presence of FAIM2 alleles and an increase in indicators of obesity, such as BMI, diastolic blood pressure, and triglycerides [33].

Genome-wide association studies (GWAS) have identified KCTD15 (potassium channel tetramerization domain containing 15) variants as being associated with increased risk of obesity. Although the detailed molecular mechanisms are not known, several lines of evidence suggest a potential role for KCTD15 in obesity, through inhibition of Wnt signaling [34].

Cluster C contains the most prominent genes described earlier in connection with the network in Figure 2b. We see the linkage of TNF (tumor necrosis factor) to adiponectin, highlighting the connection of obesity with higher risks of tumorigenesis.

Cluster D is presented by the triad of genes connected with liver glycogen synthesis, functioning of endoplasmatic proteins, and also with protein-modifiers (zinc finger 69). The R453Q and D151A variants of the H6PD gene are associated with polycystic ovarian syndrome (PCOS) and obesity, respectively. These mutations may contribute to the obesity-influenced phenotype, insulin resistance, and hyperandrogenism in the population of Caucasian women from Spain [35]. The GYS2 gene on chromosome 12p12.2 was identified in a PCOS/GWAS investigation of obesity-related conditions and has lately been confirmed by associations in an independent childhood obesity study [36].

Zfp69 encodes a transcription factor which appears to interfere with lipid storage in adipose tissue, and thereby enhances lipid deposition in the liver. In humans with type 2 diabetes, mRNA levels of the human orthologue of Zfp69 (ZNF642) were increased in adipose tissue. Thus, the transcription factor ZFP69/ZNF642 may be involved in the pathogenesis of obesity-associated diabetes [37].

The general trend of obesity as the research was captured by the data, shown in Figure 2a,b. That was determined, on the one hand, by the vectors of MeSH terms related to the problems of growing up, pregnancy and lifestyle, and on the other hand by molecular factors. The latter included the secretory role of adipocytes, a neurogenic component, as well as inflammation and tumor formation.

Further, the ScanBious functionality was applied to compose a description of each cluster in a semi-automatic mode, by copying fragments of abstracts for each cluster from Figure 2. As a result, a picture was obtained showing that clusters C and D are directly related to the general problem of obesity, cluster B may affect the nervous system and cluster A is most likely determined by genetic determinants of inheritance.

### 3.5. Relationships between the Clusters

Using clusters A–D in Figure 3 as an example, we have shown the possibilities of analyzing PubMed abstracts using the ScanBious. Information depletion was carried out: from hundreds of thousands of publications and hundreds of genes, the extremely broad problem of obesity was reduced to just a few clusters, allowed narrowing of a vast area of knowledge to a countable set of concepts reflecting development trends and key molecular actors in obesity.

The nineteen obesity-associated genes were loaded into the STRING database (v.11.0, https://string-db.org/, accessed on 22 March 2021) to search for relationships between the clusters (see Figure 3 and Figure 4a). Analysis of the results showed the relationships between gene clusters, as illustrated in the Figure 4b.

The peculiarity of the STRING is the use of various information sources such as—experimental data, co-expression, molecular interactions, and analysis of full-text publications. In view of that, one of the ScanBious clusters (Figure 3c) coincided with the STRING cluster. For the rest of analyzed objects, STRING did not establish any relationships. The only link is marked in the Figure 4 with a dashed line.

Expansion of the array of articles in the STRING made it possible to establish a relationship between clusters in the Figure 3a,d—the FBN3 (fibrillin-3) and H6PD (hexose-6-phosphate dehydrogenase). Analysis of full-text publications showed the simultaneous participation of FBN3 and H6PD in the development of reproductive and hormonal disorders. According to the OMIM (Online Mendelian Inheritance in Man) database (https://omim.org/, accessed on 22 March 2021), both genes are interrelated with the development of hormonal dysfunction and further to the polycystic ovary syndrome.

Obesity exacerbates the hormonal and clinical signs of the ovarian syndromes, and women are at a higher risk of obesity as shown by the checkmarks for pregnancy in the Figure 2a. The relationships between the clusters identified by the automatic processing of summaries require usage of additional data sources. Full-text articles and databases help to get understandable bonds between pathologies and molecular dysfunctions.

## 4. Discussion

Molecular biology creates controlled vocabularies of terms. Dictionaries include the international classification of diseases curated by the World Health Organization (WHO), the GO (gene ontology), the UniProt dictionary of genes and proteins, the annotation of chemical compounds in PubChem, and more. In the paradigm of controlled dictionaries, PubMed is in the most difficult; the object of the annotation is an abstract, a concentrated part of a scientific article. When performing genome-wide post-genomic studies, a list of genes, transcripts, and proteins may not be included in an abstract. Rather, names may be included in the appendices to the article. Medical Subject Heading experts are involved in annotating PubMed documents with a specific set of terms from the MeSH dictionary. 

We have shown that ScanBious-based MeSH correlations are relevant in consolidating data, even in a comprehensive problem such as obesity. The use of automated methods of text analysis in biology has a long history, starting with the pioneering work of Professor Barabasi [3]. According to his ideas, each biomedical term can be represented as a network node, and the edges connecting the nodes reflect the tightness of this relationship. The degree of interconnection between nodes corresponds to the co-occurrence of terms in one document in a certain context [38].

We analyzed over 100,000 obesity-related publication abstracts by downloading them from PubMed. By comparing them with the dictionary of gene names from UniProt, we obtained a collection of 622 genes that could potentially be connected with obesity. Then, data was obtained on the relationship of these genes with each other using the functionality of queries to PubMed embedded in ScanBious, which indirectly takes into account the sets of MeSH terms assigned to abstracts.

In this work, we propose a combination of the ScanBious system functionality for automatic analysis of abstracts in PubMed and a set of terms from the controlled vocabularies (UniProt, DisGeNET) to analyze the current level of knowledge on the molecular mechanisms of obesity development. Using the MeSH glossary of terms allows visualization of keywords, which index publications of the selected subject area, in the form of a semantic network. The node diameter is proportional to the depth of published studies about the issue, assessed as the number of articles (Figure 2a).

Network analysis allows users to quickly get an idea of the area under study, highlighting the most interesting fragments of the semantic map for further study. More detailed analysis, for example of the molecular mechanisms underlying obesity, is possible using a controlled vocabulary containing the names of genes and proteins. Comparison of the semantic map of key genes associated with obesity and the results of expert analysis based on the data of the DisGeNET system showed 45% of the differences in the gene lists. Along with the dictionary of genes, ScanBious can use dictionaries of drug names, diseases or methodological approaches, allowing the researcher to produce a filter for reaching the relevant group of publications.

Text-mining analysis of papers is an experimental approach to the analysis of proteomic composition, in which the experimental design includes the depletion stage of sample preparation aimed at removing highly-copied (and, therefore, uninformative) proteins and further simplifying the composition of the analyzed mixture by chromatography or two-dimensional separation. Fast fractionation will allow information about rare events that have high informational value to be obtained.

## 5. Conclusions

We propose a combination of the ScanBious functionality for automatic analysis of abstracts in PubMed and a set of terms from the controlled vocabularies (UniProt, DisGeNET). Thus we analyzed the current level of knowledge about molecular mechanisms of the obesity. The general trend of obesity as the research area was captured by the data, shown in Figure 2a,b. This field was determined, on the one hand, by the vectors of MeSH terms related to the problems of growing up, pregnancy and lifestyle, and on the other hand, by molecular factors. The latter includes the secretory role of adipocytes, a neurogenic component, as well as inflammation and tumorigenesis.

Thus, our approach allows us to identify genetic determinants, as shown in Figure 3 and Figure 4. Our findings can hardly be treated as biomarkers because ScanBious output is just a recombination of the well-known facts. ScanBious retrieves a signature which helps to distinguish different nosology-predefined situations characterized by increased body weight. From Table 1 and from the Figure 2a we also conclude that most cited authors avoid the well-known paradigms such as metabolic or lipid dysregulation.

Using MeSH terms, the PubMed search engine expands the possibilities of queries by keywords or by the last names of intensively cited authors. The PubMed screening for gene names is negatively selective to the genes of Mendelian inheritance: out of 64 genes associated with obesity, only 24 genes were identified using automated analysis. We have shown that if a query is too general, such as obesity, the answer to the query can be presented not in the form of a listing of publications, but rather in the form of graphical relationships between terms; in particular genes or MeSH terms highlighted the key concepts of the problem. Using ScanBious for this purpose, we came to the conclusions below.

(1) Published knowledge on the problem of obesity is hidden under a plethora persistent terms referring both to the genes and proteins that are most known in the pathogenesis of diseases, and to typical states of human development: pregnancy, growing up, lifestyle, diet, and so forth.

(2) The combination of the DisGeNET expert knowledgebase with the PubMed-driven processing of abstracts in ScanBious allowed us to shrink the huge field to a limited number of relationships (Figure 3) highlighting the somewhat underestimated players of inflammation, glucose regulation, and heredity as intrinsic aspects of obesity.

## Figures and Tables

**Figure 1 jpm-11-00246-f001:**
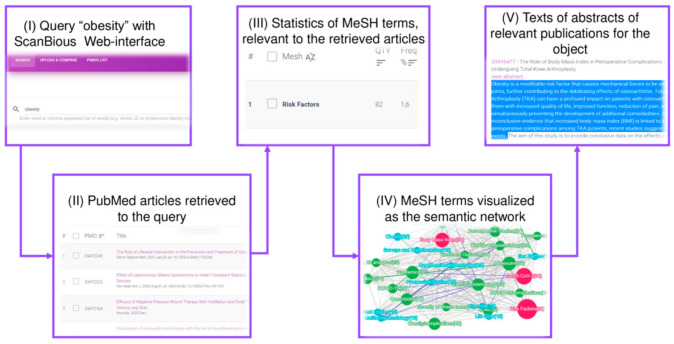
ScanBious Web-interface. (**I**) Query “obesity” with ScanBious Web-interface, (**II**) PubMed articles retrieved to the query, (**III**) statistics of the MeSH terms, relevant to the retrieved articles, presented as a network of MeSH terms, (**IV**) MeSH terms visualized as the semantic network, (**V**) texts of abstracts of relevant publications for the object.

**Figure 2 jpm-11-00246-f002:**
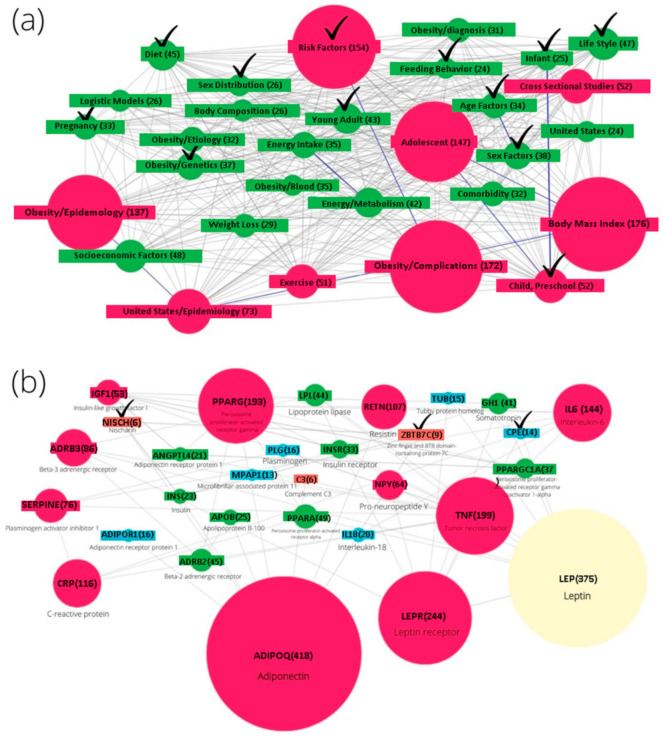
(**a**) Obesity at-a-glance. The compendium of PubMed abstracts from the top ten most cited authors is depicted as a network of MeSH terms. The size of the nodes reflects the occurrence of the MeSH terms in the sampled abstracts. (**b**) Gene network based on DisGeNET data obtained on the request “obesity.” The publications from DisGeNET were retrieved as a file in which the names of genes were assigned to the PubMed identifiers, and in this format were uploaded to ScanBious for visualization and interactive work with the texts of the abstracts of publications. The number of publications in which the name of the gene was found is indicated in parentheses after the gene name. The checkmarks indicate the notes of the network which are described in the text.

**Figure 3 jpm-11-00246-f003:**
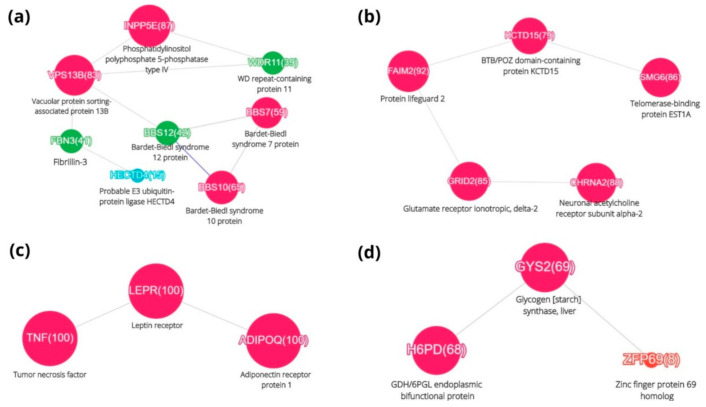
Four clusters of the obesity-relevant semantic network. Clusters (**a**–**d**) were obtained as a result of comparing data from the DisGeNET expert system and the results of automatic processing of abstracts of scientific publications in ScanBious, with a threshold value of the Jaccard index > 0.2, with the condition that there were at least five articles for the object being visualized as a network node.

**Figure 4 jpm-11-00246-f004:**
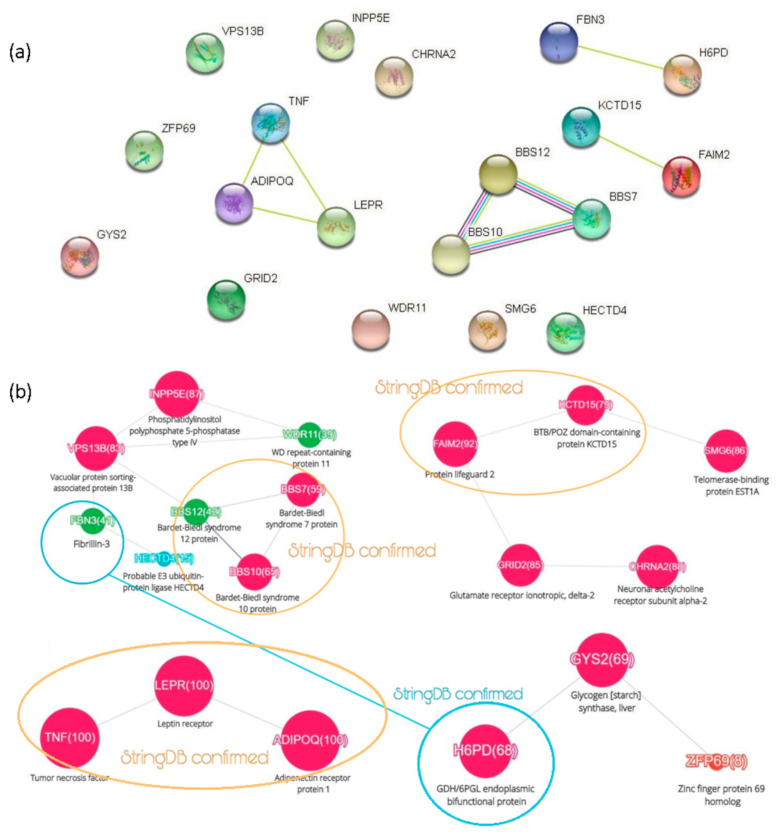
The relationship among the clusters as found by using STRING (**a**) with consequent reproduction of the Figure 3 with graphical illustration of the links between genes (**b**).

**Table 1 jpm-11-00246-t001:** Top five genes related to the Mendelian forms of obesity, selected according to the number of gene-associated PubMed abstracts. N.Diseases and N.PMIDs were retrieved as a result of the DisGeNET search for a given gene, and denote the numbers of diseases and PubMed identifiers, respectively.

Gene Name ^1^	Number of References			Protein Name
	Obesity/PubMed ^2^	N.Diseases	Obesity/N.PMIDs	
FTO	26	286	426	Alpha-ketoglutarate-dependent dioxygenase
POMC	22	873	97	Proopiomelanocortin
MC4R	17	149	283	Melanocortin receptor 4
LEPR ^3^	13	416	214	Leptin receptor
BDNF	8	992	88	Brain-derived neurotrophic factor

^1^ collected from the reviews [1,20,21]. ^2^ sorted by this column. ^3^ appeared in the DisGeNET network (Figure 2b).

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
