# Peer review of "ScanBious: Survey for Obesity Genes Using PubMed Abstracts and DisGeNET"

_jpm, 2021, doi:10.3390/jpm11040246_

Round 1
Reviewer 1 Report
The manuscript of “ScanBious: Survey for Obesity Genes using PubMed Abstracts and DisGeNET” is described to integrate information of human gene - obesity associations from various data of DisGeNET platform and PubMed abstracts. ScanBious interface was used to sort by frequency of occurrence of key concepts and research objects in a given subject area. Overall, this manuscript is well-written, the content is clear and explained in detail.
Based on your result “3.5 Combining the PubMed Survey with DisGeNET Data on Obesity Genes”, it seems like the relationship of genes and obesity is easier for audient, but will this possible to display the relationship among each cluster as shown in figure 3 (a) - (d)? In figure 2, you need to describe what does the black marked checkmark indicates in the legend. In the manuscript, there is a missing result 3.4.
Author Response
Dear Reviewer, thank you very much for your time and valuable feedback on the manuscript. We carefully read the comments and in accordance with them contributed to the article by a number of edits, namely:
- Based on your comment we expanded the manuscript to display the relationship among clusters that are shown in Figure 3. We added section “3.5. Relationships between the Clusters” and introduced Figure 4.
- We described what the black checkmark indicates in the legend of Figure 2.
- “In the manuscript, there is a missing result 3.4”. Corrected, sorry for the mistake.
Respectfully yours,
Svetlana Tarbeeva,
Corresponding Author
Reviewer 2 Report
Obesity is a serious health crisis in developed countries as well as in worldwide. It is important to identify biomarker that can helps to program personalized medicine. Tarbeeva et all use ScanBious web tool to identify signature that could be used for personalized care.
- Line #53 author need to separate SARS-COVID19 terminology. SARS-CoV-2 is a virus that causes COVID-19
- Obesity is a disease of metabolic and lipid dysregulation and author did not talk about this areas in the manuscript which must be addressed.
- Conclusion section need to bring the major finding bioamrkers and talk more about their potential in obesity care.
Author Response
Dear Reviewer,
Thank you very much for your criticism and valuable feedback on the manuscript. We carefully read the comments and in accordance with them corrected the article by a number of edits:
“Line #53 author need to separate SARS-COVID19 terminology. SARS-CoV-2 is a virus that causes COVID-19”
corrected
“Obesity is a disease of metabolic and lipid dysregulation and author did not talk about this areas in the manuscript which must be addressed”.
We have addressed this issue in the Introduction, Result and Conclusion sections. We are very grateful for your comment as we see that ScanBious neglects not only the high-abundant terms but also the interlinked concepts like metabolic and lipid dysregulation. That is already on the top of our future research and development of ScanBious.
“Conclusion section need to bring the major finding biomarkers and talk more about their potential in obesity care”.
In the Conclusion section we have described the perspective of our findings to become biomarkers. We highlighted that our findings have no any attitude to the biomarkers.
Respectfully yours,
Svetlana Tarbeeva,
Corresponding Author
Reviewer 3 Report
Paper by Tarbeeva et al. propose a combination of the ScanBious system functionality for automatic analysis of abstracts in PubMed and a set of terms from the controlled vocabularies (UniProt, DisGeNET) to analyze the current level of knowledge on the molecular 357 mechanisms of obesity development.
In particular Authors have presented convincing evidences that combination of the DisGeNET expert knowledgebase with the PubMed-driven processing of abstracts in ScanBious might allow to highlight the most relevant relationships among genes involved and avoid to underestimate and evaluate key genes missed using ScanBious alone.
This approach appear very promising to dissect acquired knowledge and focalize the most relevant genes involved in a multifactorial disease as obesity and proposed methodology might be easily applied to other multifactorial disease from metabolic syndrome to atherosclerosis to cancer.
Author Response
Dear Reviewer,
Thank you very much for your time and valuable feedback on the manuscript. We are grateful for such a high assessment and we hope that the described method will be able to help many researchers from various fields of medical sciences.
Respectfully yours,
Svetlana Tarbeeva,
Corresponding Author
Round 2
Reviewer 1 Report
The authors address the concerns and questions properly and accordingly, the revised version of the manuscript seems to be well written and should provide useful results. I recommend the revised manuscript for publication without further revision. The authors' hard work and patience in revising the manuscript are truly appreciated.